# Association between Mean Nocturnal Baseline Impedance (MNBI) and Post-Reflux Swallow-Induced Peristaltic Wave Index (PSPW) in GERD Patients

**DOI:** 10.3390/diagnostics13243602

**Published:** 2023-12-05

**Authors:** Elena Roxana Sararu, Razvan Peagu, Carmen Fierbinteanu-Braticevici

**Affiliations:** 1Internal Medicine II and Gastroenterology Department, Emergency University Hospital Bucharest, Carol Davila University of Medicine and Pharmacy, 050474 Bucharest, Romania; 2Internal Medicine Department, Sanador Hospital, 010991 Bucharest, Romania

**Keywords:** GERD, MNBI, PSPW, reflux hypersensitivity, functional heartburn

## Abstract

Gastroesophageal reflux disease (GERD) is one of the most common gastrointestinal disorders in the world. Two parameters, mean nocturnal baseline impedance (MNBI) and post-reflux swallow-induced peristaltic wave index (PSPW), have been recently proposed to help differentiate GERD phenotypes. Our study aimed to assess whether there is any correlation between the two parameters, while also taking a look at their ability to distinguish between GERD phenotypes. We recruited 81 patients who were divided into 4 groups based on their GERD phenotype: erosive reflux disease (ERD), non-erosive reflux disease (NERD), reflux hypersensitivity (RH), and functional heartburn (FH). Both MNBI (AUROC 0.855) and PSPW (AUROC 0.835) had very good performances in separating ERD patients from non-ERD patients. PSPW (AUROC 0.784) was superior to MNBI (AUROC 0.703) in distinguishing NERD patients from patients with RH or FH. The PSPW index (AUROC 0.762) was more effective than MNBI (AUROC 0.668) in separating RH from FH. We found that PSPW and MNBI have a strong statistical correlation (Pearson correlation coefficient, r = 0.722, *p* < 0.001). Furthermore, PSPW predicted pathological MNBI (<2292 Ω) with good performance (AUROC 0.807). MNBI and PSPW are useful in distinguishing GERD phenotypes, with a strong correlation between the two parameters.

## 1. Introduction

Gastroesophageal reflux disease (GERD) is one of the most common gastrointestinal disorders in the world. According to the Montreal Consensus of 2006, this condition consists of the retrograde passage of gastric acid into the esophagus, which can lead to the onset of reflux symptoms [1]. 

Recent data show that GERD affects as many as 1 billion individuals worldwide [2,3]. The overall prevalence of GERD worldwide is estimated to be around 14%, with significant geographical variations [2]. North America has the highest prevalence, with an estimated 19.55%, while Latin America has the lowest, with an estimated 12.88%, while Europe’s prevalence falls at around 14% [2]. Patients with a lower income have a significantly higher prevalence of GERD (11.69%) than those with a medium income (8.42%) and those with a higher income (7.68%) [2].

Given the high prevalence, GERD is considered to be one of the most expensive digestive diseases due to frequent visits to the doctor, the cost of proton pump inhibitors, and the cost of testing and diagnosis. The cost is estimated to be approximately 760 million pounds sterling per year in the United Kingdom, and 24 billion dollars per year in the United States [2]. 

The increased incidence of GERD may be explained by the increase in living standards, the rising rates of obesity worldwide, disorganized lifestyles, diets rich in fat and fast food, carbonated and alcoholic beverages, and the use of spicy or sour foods or products such as chocolate or coffee in excess [2]. It has also been observed that there is a higher incidence rate of reflux disease in men, with men often presenting with erosive esophagitis, while women are more prone to non-erosive reflux disease [4]. 

Multichannel ambulatory impedance and pH monitoring (MII-pH) is a frequently used procedure in the clinical practice of gastroenterologists for the diagnosis and monitoring of gastroesophageal reflux episodes. Among the most commonly used parameters in this investigation are: total acid exposure time (AET), total number of reflux episodes recorded over 24 h, probability of symptom association with reflux episodes (SAP), and symptom index (SI). 

One of the challenges in GERD patients is properly diagnosing the many GERD phenotypes. According to Rome IV, GERD phenotypes are defined as erosive reflux disease (ERD), non-erosive reflux disease (NERD), functional heartburn (FH), and reflux hypersensitivity (RH) [5]. ERD is characterized by abnormal endoscopy findings for erosions with abnormal acid exposure [5]. NERD is defined by normal endoscopy findings with abnormal acid exposure, with either positive or negative symptom–reflux association [5]. FH is characterized by normal acid exposure to the esophagus, but with a positive symptom–reflux association and normal endoscopy [5]. RH is defined by abnormal acid exposure to the esophagus, but with a negative symptom–reflux association and normal endoscopy [5].

In recent years, there have been attempts to identify and propose new parameters to facilitate the diagnosis of GERD phenotypes [6]. Thus, two parameters have been identified, mean nocturnal baseline impedance (MNBI) and post-reflux swallow-induced peristaltic wave index (PSPW), which may be useful in differentiating gastroesophageal reflux disease phenotypes, such as ERD, NERD, FH, and RH [6,7]. MNBI represents the mean baseline impedance determined from MII-pH data collected over three 10 min intervals during the night (around 1:00 AM, 2:00 AM, and 3:00 AM) [8]. PSPW evaluates the effectiveness of the esophagus’s ability to clear harmful substances. This clearance process (chemical clearance) involves transporting salivary bicarbonate and epidermal growth factor to the lower part of the esophagus, which aids in repairing mucosal damage caused by acid reflux [9].

These two parameters have shown promise in predicting treatment response to acid suppression therapy [10,11,12]. These findings are not fully understood, but one possible explanation is that MNBI correlates with esophageal mucosal integrity in histological assessments [13]. More specifically, MNBI values have shown to be associated with the presence of esophageal mucosal dilated intercellular spaces (DIS), which is one the most sensitive markers of esophageal mucosal damage [13,14,15]. The exact cause of DIS is not fully understood, but it is thought to be caused by damage to the esophageal lining from acid reflux. When acid reflux occurs, it can break down the tight junctions between the cells of the esophageal lining, allowing fluid and other substances to pass more easily between the cells [14,15,16]. The presence of dilated intercellular spaces is not exclusive to patients with erosive esophagitis; it is also observed in patients with symptomatic non-erosive reflux disease [17].

Considering the comparable applications of MNBI and PSPW, the present study sought to investigate the potential correlation between these two parameters and evaluate their efficacy in distinguishing between GERD phenotypes.

## 2. Materials and Methods

We conducted a retrospective study of 81 patients with heartburn from March 2021 to February 2023 at the Gastroenterology Department of the Bucharest Emergency University Hospital in Romania. The study was conducted in accordance with the Declaration of Helsinki and was approved by the local ethics committee. All patients provided informed written consent before participating in the study.

We included patients over the age of 18 who had typical symptoms of GERD (heartburn and acid regurgitation) at least 2 times per week in the last 6 months. All patients underwent a complete medical exam, including a detailed medical history, dietary habits, and gastroesophageal reflux disease questionnaire (GERDQ). We excluded patients with a history of heavy alcohol use, esophageal motility disorders, surgery to the esophagus or stomach, esophageal or gastric tumors, esophageal varices, stroke, severe neurological conditions, or otolaryngologic disorders. Patients were instructed to stop PPI and histamine H2 antagonist usage 2 weeks prior to MII-pH testing. 

All patients underwent upper digestive endoscopy and MII-pH testing, and some patients underwent esophageal manometry to rule out motility disorders. Endoscopy was performed by 2 specialists who were blinded to the results of the MII-pH tests. The severity of erosive reflux disease was ranked using the Los Angeles (LA) classification. 

All patients received PPI therapy for 8 weeks after investigations and the response to PPI helped to define the GERD phenotypes. Response to PPI was defined as at least a 50% improvement in esophageal reflux symptoms on follow-up evaluation based on questionnaire data from the GERDQ. Patients who had inconclusive or negative findings for GERD based on upper endoscopy, MII-pH, or response to PPI therapy were excluded from the study.

All patients underwent multichannel intraluminal impedance and pH monitoring using the Ohmega Ambulatory Impedance pH Recorder device (Laborie, Enschede, The Netherlands). The MII-pH device is equipped with a catheter probe containing six impedance sensors and two pH sensors, which was calibrated in buffer solutions with pH of 4.0 and 7.0, and after that was inserted transnasally to the level of the lower esophageal sphincter (LES). The pH sensor was placed at 5 cm above the lower esophageal sphincter. The six intraluminal impedance sensors are located at 3, 5, 7, 9, 15, and 17 cm from the lower esophageal sphincter, respectively Z6 = 3 cm, Z5 = 5 cm, Z4 = 7 cm, Z3 = 9 cm, Z2 = 15 cm, and Z1 = 17 cm. To correctly evaluate the catheter’s positioning above the lower esophageal sphincter, upper gastrointestinal endoscopy was performed prior to MII-pH in all patients included in the study. Patients were instructed to stop acid-lowering medication 2 weeks prior to MII-pH. Patients were taught how to use the event buttons to record meal times, body positions (supine or upright), and symptom episodes. The recording lasted for approximately 24 h, and the data were manually analyzed by two physicians using Medical Measurement System database software program (Medical Measurement System B.V., Version 9.3 build 2634) provided by Laborie. Total acid exposure time (AET) was defined as the percentage of time that the pH of the distal esophagus was below 4.0 and was calculated by dividing the total time that the pH was below 4.0 by the total monitoring time, excluding meal times. A total AET of 6% or higher was considered abnormal, less than 4% was considered physiologic, and values between 4% to 6% were considered borderline or inconclusive [6].

Patients were classified into four GERD phenotypes: erosive reflux disease (ERD), non-erosive reflux disease (NERD), reflux hypersensitivity (RH), and functional heartburn (FH) based on the Lyon Consensus and Rome IV criteria [6,18]. ERD was diagnosed in patients who had esophagitis LA grade C or D, peptic strictures or Barrett’s esophagus on endoscopy, and abnormal AET (>6%) that responded to PPI therapy. The NERD group was defined by an abnormal AET (>6%) with negative endoscopy findings that responded to PPI treatment. Patients with normal endoscopy findings, a normal AET (<4%), and a positive symptom association probability (SAP) were classified as RH. Patients with a negative SAP, normal AET (<4%), and no endoscopy abnormalities were defined as FH.

MNBI was measured from the most distal impedance channel during nighttime with the patients in a recumbent position. MNBI was calculated from the mean value from 3 distinct timeframes at 1:00 AM, 2:00 AM, and 3:00 AM across stable nocturnal 10 min periods (swallows, pH drops, and refluxes during this period were excluded) [19]. This parameter was measured in ohms (Ω). 

PSPW represents the primary esophageal mechanism for chemical clearance and was defined as a 50% decrease in impedance that occurred within 30 s of a reflux event, starting at the most proximal impedance channel and reaching the most distal impedance channel, followed by at least a 50% return to baseline. The PSPW index was calculated by dividing the number of PSPWs by the number of reflux events [20].

### Statistical Analysis

IBM SPSS 26 (Statistical Package for the Social Sciences Inc., IBM corporation, Armonk, NY, USA) and Microsoft Office (Microsoft Corporation, One Microsoft Way Redmond, Washington, DC, USA) were used to analyze the data. The median and interquartile range were used to summarize the parameter values. To compare the parameter values between two distinct groups, the Mann–Whitney U test, a non-parametric test, was implemented. One-way ANOVA analysis was used when comparing multiple groups. To account for multiple comparisons and maintain the integrity of the statistical analysis, the Bonferroni correction was applied. The strength and direction of the linear relationship between the PSPW index and MNBI were evaluated using Pearson’s correlation coefficient. Pathologic values for MNBI were defined in our study as values under 2292 Ω based on other European studies [21,22]. To assess the diagnostic performance of parameters such as MNBI and PSPW, the area under the ROC curve (AUROC) analysis was employed. The cut-off values were selected based on the ROC curve with the optimal sensitivity and specificity, and the best overall diagnostic performance. A *p*-value of less than 0.05 was considered statistically significant.

## 3. Results

A total of 81 patients were recruited for our study. GERD phenotype characteristics are presented in Table 1. Of the patients included in the study, 35 (43.2%) had erosive reflux disease (ERD), 12 (14.8%) had non-erosive reflux disease (NERD), 16 (19.8%) had reflux hypersensitivity (RH), and 18 (22.2%) had functional heartburn (FH). GERD phenotype characteristics are presented in Table 1. Our study was composed of 41 male patients (50.6%) and 40 female patients (49.4%). There was no correlation between the age, sex, or BMI of the patients across the GERD phenotypes. 

There was a significant statistical difference between MNBI values between GERD phenotypes (*p* = 0.001) presented in Figure 1. ERD patients had the lowest values of MNBI, while patients with FH had the highest values. There were wider ranges of MNBI results in patients with ERD and NERD when compared with patients with RH and FH. 

When it came to PSPW values across GERD phenotypes, the differences were much more obvious (Figure 2). Patients with ERD had the lowest PSPW values, followed by higher values, in order, by NERD, RH, and FH, which had the highest values. 

Our study evaluated the performance of PSPW and MNBI in distinguishing between GERD phenotypes (Table 2). Using a receiver operating characteristic (ROC) curve analysis, we assessed the ability of PSPW and MNBI to distinguish ERD from non-erosive reflux phenotypes (NERD, RH, and FH). We also evaluated the performance of PSPW and MNBI in discriminating between NERD and functional phenotypes (RH and FH), between RH and FH, and between FH and non-FH phenotypes (ERD, NERD, and RH).

Both MNBI (AUROC 0.855, CI 95% 0.780–0.930) and PSPW (AUROC 0.835, CI 95% 0.765–0.931) had very good performances in separating ERD patients from non-ERD patients (NERD, RH, and FH), with MNBI showing slightly better results. A MNBI cut-off value of 1818 Ω was able to predict the presence of ERD with 88% sensibility and 79% specificity, while for PSPW, a cut-off value of 37% showed 80% sensibility and 70% specificity. 

PSPW (AUROC 0.784, CI 95% 0.650–0.919) was superior to MNBI (AUROC 0.703, CI 95% 0.583–0.824) in distinguishing NERD patients from patients with RH or FH, with both metrics demonstrating acceptable performance. A cut-off value of 47.5% for PSPW was able to diagnose the presence of NERD with 77% sensibility and 83% specificity. MNBI showed poorer performance, a cut-off value of 1975 Ω showed a sensibility of 73% and specificity of 59% in the diagnostic of NERD.

For separating RH patients from FH patients, PSPW (AUROC 0.762, CI 95% 0.621–0.924) was more effective than MNBI (AUROC 0.668, CI 95% 0.482–0.855). A cut-off value of 54.5% for PSPW was able to differentiate RH from FH with a sensibility of 83% and a specificity of 50%. MNBI had poor performance in distinguishing RH from FH, a cut-off value of 2164 Ω showed 72% sensibility and 50% specificity. 

When it came to the diagnosis of functional heartburn (FH), both parameters showed good performances, but PSPW (AUROC 0.871, CI 95% 0.778–0.965) proved to be superior to the MNBI (AUROC 0.821, CI 95% 0.727–0.914). A PSPW cut-off value of 54.5% was able to diagnose FH with a sensibility of 83% and a specificity of 83%. Meanwhile, MNBI was able to predict the presence of FH for a cut-off value of 2280 Ω with 72% sensibility and 80% specificity. 

Given that both MNBI and PSPW have similar applications and performance in diagnosing GERD phenotypes, we evaluated if there was any correlation between the two parameters. Firstly, we used the Pearson correlation coefficient, which showed significant statistical correlation between MNBI and PSPW (r = 0.722, *p* = 0.001). 

We also investigated whether PSPW could predict the presence of pathological MNBI (defined as MNBI < 2292 Ω based on other European studies) [22]. PSPW was able to diagnose pathological MNBI with an excellent area under the ROC curve of 0.807 (CI 95% 0.705–0.904) (Figure 3). For a cut-off value of 47%, it was able to predict the presence of pathological MNBI with 81% sensitivity and 79% specificity.

## 4. Discussions

GERD is one of the most common gastroesophageal diseases in the world, with a recent global study reporting the estimates for GERD at 783 million prevalent cases and 309 million incident cases, with over 6.0 million years lived with GERD disability, with prevalence increasing over the past three decades [3]. Interestingly, countries with lower levels of economic development experience higher costs associated with GERD [3]. Given the high prevalence and financial burden of GERD, accurate diagnosis of GERD phenotypes is essential. However, diagnosing GERD phenotypes remains a challenge. Our study investigated whether two new parameters (MNBI and PSPW) are useful for the diagnosis of GERD phenotypes and whether there was any correlation between the two parameters. 

Our investigation found that both PSPW and MNBI are effective methods for differentiating between all major GERD phenotypes. Firstly, PSPW (AUROC 0.835) and MNBI (AUROC 0.855) were effective in distinguishing ERD from non-ERD phenotypes. Secondly, our research also showed good results for MNBI (AUROC 0.703) and PSPW (AUROC 0.784) in differentiating NERD from RH and FH. Thirdly, we found that both MNBI (AUROC 0.871) and PSPW (AUROC 0.821) were accurate in separating FH patients from those with ERD, NERD, and RH. Lastly, PSPW (AUROC 0.762) was superior to MNBI (AUROC 0.668) in distinguishing RH and FH from each other.

Given that both MNBI and PSPW have similar applications, we found that PSPW and MNBI have a strong statistical correlation (Pearson correlation coefficient, r = 0.722, *p* < 0.001). Our study also found that PSPW was also strongly correlated with pathological MNBI values (<2292 Ω). The pathological MNBI value of 2292 Ω was established by Frazzoni et al. in a multicentric European study in 2016 [19]. PSPW was able to predict pathological MNBI values with good performance (AUROC 0.807), with a cut-off value of 47% that achieved 81% sensitivity and 79% specificity. Slightly better results were found in another European study by Ribolsi et al., who found that a PSPW cut-off value of 53% predicted pathological MNBI (<2292 Ω) with a sensitivity 88%, specificity 86.4% [21]. This is an important finding, as pathological MNBI and the PSPW index appear to be significantly associated with PPI therapy response in the European population [9,10,23]. It is important to note that the pathological MNBI values used in our study are based on other European studies, however the definition of pathological MNBI by cut-off value seems to vary by geographical region, with lower values observed in Asian populations, for example [13,24]. Possible explanations for this correlation may be due to the fact that effective chemical clearance (measured by PSPW), neutralizes esophageal acid by increasing esophageal pH, thus serving as a critical defense mechanism by preserving esophageal mucosal integrity (measured by MNBI) [25]. Histopathological studies assessing mucosal integrity through dilated intercellular spaces (DIS) have demonstrated a correlation between MNBI values and mucosal integrity [13,26,27].

FH is one of the most common causes of PPI treatment failure. Therefore, it is important to make a correct diagnosis, as treatment for FH differs from other types of GERD phenotypes. Neuromodulators, such as tricyclic antidepressants (TCAs), selective serotonin reuptake inhibitors (SSRIs), serotonin norepinephrine reuptake inhibitors (SNRIs), as well as psychological intervention, are the mainstay of treatment for FH [28]. Our study showed that both PSPW and MNBI are accurate methods for differentiating FH patients from other phenotypes (ERD, NERD, RH) with AUROC of 0.871 and 0.821, respectively. Similar findings have been reported by other studies [22,29]. For example, Sun et al. found that distal MNBI (AUROC 0.721) and PSPW (AUROC 0.779) were both effective methods of separating FH from ERD, NERD, and RH [29]. In a study comparing FH and ERD patients, Frazzoni et al. found that both PSPW (AUROC 0.866) and MNBI (AUROC 0.838) were able to distinguish the two phenotypes [22]. They also found that PSPW (AUROC 0.866) was superior to MNBI (AUROC 0.677) in separating NERD from FH [22]. When it comes to separating FH from NERD, Yoshimine et al. (AUROC 0.73) and Tenca et al. (AUROC 0.960) had better results using MNBI [30,31]. 

One of the main challenges of diagnosing FH is separating it from RH. In our study, MNBI showed poor accuracy in differentiating FH from RH (AUROC 0.668), while PSPW had better accuracy (AUROC 0.762). One study conducted in Asia showed similar results to ours, where PSPW (AUROC 0.728) had better diagnostic accuracy than MNBI (AUROC 0.643) [32]. Frazzoni et al. had better results when they analyzed the results from a European cohort, which showed excellent accuracy for both PSPW (AUROC 0.924) and MNBI (AUROC 0.864) in distinguishing FH from RH [33]. Symptom association probability (SAP) and symptom index (SI) have traditionally been the parameters most commonly used to separate RH from FH. However, the clinical value of SAP and SI has recently been questioned, particularly when discordant results are found [33]. One study by Frazzoni et al. demonstrated that both PSPW and MNBI were able to predict the presence of RH with higher accuracy than SAP and SI [33]. When there is uncertainty in diagnosing GERD, particularly when the AET is normal, SAP and SI are negative or inconsistent, or the patient acknowledges poor symptom tracking, the PSPW index and MNBI can prove valuable parameters to assess. Further research is warranted to determine whether these conventional parameters (SAP and SI) can be effectively replaced by PSPW or MNBI.

The most important issue to take into account is that both MNBI and PSPW values vary widely across geographic locations, with large discrepancies between what is considered normal in different regions of the world [24]. This is likely due to a number of factors, including genetics, diet, and culture [24]. For example, higher MNBI values were observed in Asia and South Africa compared to the rest of the world. Differences were also found with PSPW values, which were greater in Western countries, with lower values present in Asia and South Africa [24]. It is important to note that these are just general trends, and there is a great deal of individual variation within each region. Furthermore, the device used to record MII-pH needs to be taken into account, as different thresholds are seen based on the manufacturer of the device [24]. In a study comparing 126 Laborie tracings to 265 Diversatek tracings, the Laborie system (the device used in our study as well) showed higher MNBI values than the Diversatek system [24]. The variations in MNBI and PSPW values obtained using different MII-pH measuring devices could be attributed to the distinct characteristics of the catheters, the types of pH electrodes and impedance electrodes employed, or the differences in the amplifiers or software algorithms incorporated into the devices [24].

Given that both geographical region and MII-pH device seem to influence MNBI and PSPW values, appropriate cut-offs that take into account these variables need to be considered. This ultimately may prove cumbersome in the future when trying to compose proper guidelines. 

For future perspectives, we have to take into account artificial intelligence (AI), which is a rapidly developing field with the potential to revolutionize the diagnosis and treatment of many diseases, including gastroesophageal reflux disease. AI algorithms can be trained on large datasets of patient data to learn complex patterns and relationships that may be difficult for human experts to discern. This ability to learn from data makes AI well suited for the task of measuring novel pH impedance metrics, such as MNBI and PSPW, which are complex and prone to human error. Implementing software updates by MII-pH manufactures that automatically calculate MNBI and PSPW values with high accuracy could also be a more cost-effective strategy.

Our study had some limitations. Firstly, it was a retrospective analysis performed at a single medical center, which may have introduced selection bias. Secondly, we had a small sample size and did not include healthy patients or patients taking proton pump inhibitors. Lastly, we calculated MNBI and PSPW manually, as no software was available at the time, and this may have introduced human error into the equation. Our study’s strength lies in the simultaneous measurement of MNBI and PSPW across multiple GERD phenotypes, while also revealing a robust correlation between these two parameters.

## 5. Conclusions

Our research found a strong correlation between PSPW and MNBI, as both parameters were useful in distinguishing between GERD phenotypes. These findings are significant because it could lead to the development of new and more accurate methods for diagnosing and classifying GERD patients. MNBI and PSPW values are easy to extract from MII-pH studies, without additional costs, but the lack of formal software can make data collection prone to human error. Additionally, their values vary depending on the measuring apparatus and geographical location [24]. Further studies are needed to establish proper cut-off values based on phenotype, geographical region, and device used.

## Figures and Tables

**Figure 1 diagnostics-13-03602-f001:**
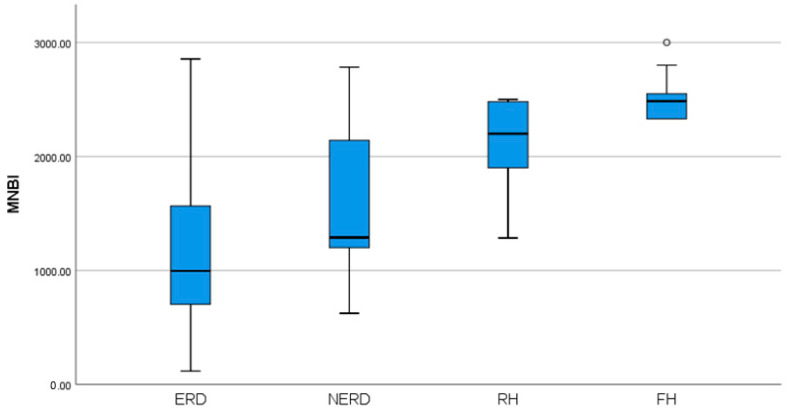
Distribution of MNBI values between GERD phenotypes (*p* = 0.001). MNBI = mean nocturnal baseline impedance, ERD = erosive reflux disease, NERD = non-erosive reflux disease, RH = reflux hypersensitivity, FH = functional heartburn.

**Figure 2 diagnostics-13-03602-f002:**
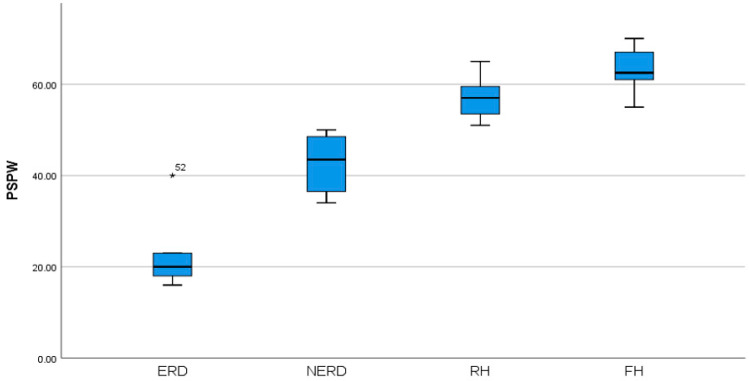
Distribution of PSPW values between GERD phenotypes (*p* = 0.001). PSPW = post-reflux swallow-induced peristaltic wave index, ERD = erosive reflux disease, NERD = non-erosive reflux disease, RH = reflux hypersensitivity, FH = functional heartburn, * = PSPW value of patient number 52 which had abnormal results.

**Figure 3 diagnostics-13-03602-f003:**
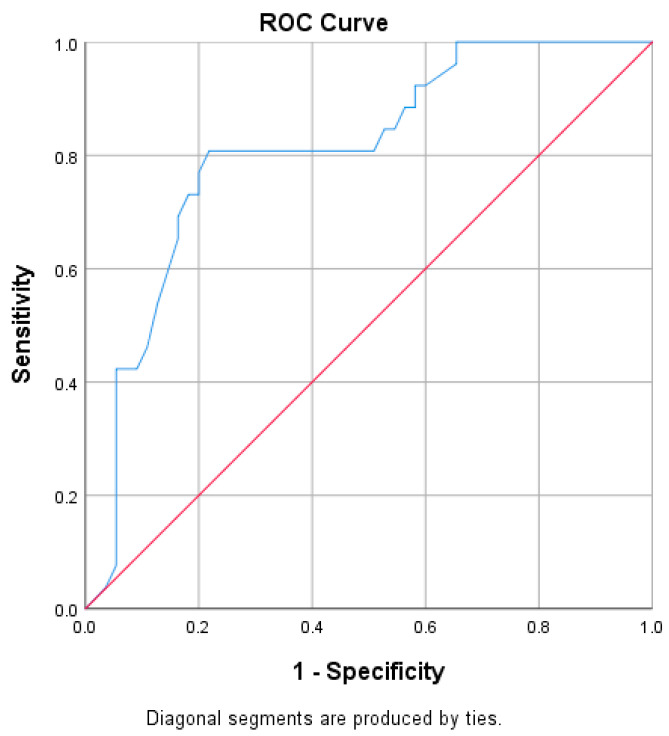
ROC curve of PSPW in the diagnosis of pathologic MNBI < 2292 Ω (AUROC 0.807).

**Table 1 diagnostics-13-03602-t001:** Characteristics of patient population (*n* = 81).

	ERD (*n* = 35)	NERD (*n* = 12)	RH (*n* = 16)	FH (*n* = 18)	*p* Value
**Age (years)**	53.2 ± 13.7	47.6 ± 9.4	51.5 ± 9.38	46.8 ± 12.4	0.315
**Sex**	Male = 14	Male = 8	Male = 10	Male = 9	0.610
	Female = 21	Female = 4	Female = 6	Female = 9	
**BMI (kg/m^2^)**	23.8 ± 2.3	25 ± 4	24.6 ± 2.9	23.8 ± 2.5	0.613
**AET (%)**	9.5 ± 2.4	8 ± 1.5	2.35 ± 1.62	1.4 ± 2.1	0.001
**MNBI (Ω)**	1159 ± 679	1694 ± 738	2147 ± 356.5	2350 ± 390	0.001
**PSPW (%)**	28.7 ± 14.4	41.2 ± 7	47.9 ± 13.9	56 ± 12.8	0.001

BMI = body mass index, AET = acid exposure time, MNBI = mean nocturnal baseline impedance, PSPW = post-reflux swallow-induced peristaltic wave index, ERD = erosive reflux disease, NERD = non-erosive reflux disease, RH = reflux hypersensitivity, FH = functional heartburn.

**Table 2 diagnostics-13-03602-t002:** Performance of PSPW index and MNBI in distinguishing GERD phenotypes.

**ERD vs. Non-ERD (NERD, RH, and FH) Phenotypes**					
	AUROC (CI 95%)	Cut-off	Sensibility	Specificity	*p* value
**MNBI**	0.855 (0.780–0.930)	1818 Ω	88%	79%	0.001
**PSPW**	0.835 (0.765–0.931)	37%	80%	70%	0.001
**NERD vs. Functional phenotypes (RH and FH)**					
	AUROC (CI 95%)	Cut-off	Sensibility	Specificity	*p* value
**MNBI**	0.703 (0.583–0.824)	1975 Ω	73%	59%	0.038
**PSPW**	0.784 (0.650–0.919)	47.5%	77%	83%	0.001
**RH vs. FH**					
	AUROC (CI 95%)	Cut-off	Sensibility	Specificity	*p* value
**MNBI**	0.668 (0.482–0.855)	2164 Ω	72%	50%	0.095
**PSPW**	0.762 (0.621–0.924)	54.5%	83%	50%	0.008
**FH vs. non-FH (ERD, NERD and RH) phenotypes**					
	AUROC (CI 95%)	Cut-off	Sensibility	Specificity	*p* value
**MNBI**	0.821 (0.727–0.914)	2280 Ω	72%	80%	0.001
**PSPW**	0.871 (0.778–0.965)	54.5%	83%	83%	0.001

MNBI = mean nocturnal baseline impedance, PSPW = post-reflux swallow-induced peristaltic wave index, ERD = erosive reflux disease, NERD = non-erosive reflux disease, RH = reflux hypersensitivity, FH = functional heartburn, CI = confidence interval.

## Data Availability

The data presented in this study are available on request from the corresponding author. The data are not publicly available due to privacy and ethical restrictions.

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
