# Peer review of "Association between Mean Nocturnal Baseline Impedance (MNBI) and Post-Reflux Swallow-Induced Peristaltic Wave Index (PSPW) in GERD Patients"

_diagnostics, 2023, doi:10.3390/diagnostics13243602_

Round 1

Reviewer 1 Report

Comments and Suggestions for Authors

minor changes needed:

Introduction

- Provide more details on the prevalence and burden of GERD globally and LPR. Cite key statistics.

- Elaborate on the different phenotypes of GERD - erosive reflux disease (ERD), non-erosive reflux disease (NERD), functional heartburn (FH), reflux hypersensitivity (RH).

- Explain the pathophysiology behind altered esophageal mucosal integrity in GERD. Discuss dilated intercellular spaces (DIS).

- Introduce multichannel intraluminal impedance pH monitoring. Define key parameters like mean nocturnal baseline impedance (MNBI) and post-reflux swallow-induced peristaltic wave index (PSPW).

- Discuss existing literature on the utility of MNBI and PSPW in diagnosing GERD phenotypes. 

- State the rationale and objectives of your study clearly.

Methods

- Provide details on study design, setting, timeline and ethical approvals obtained.

- Explain patient recruitment strategy and inclusion/exclusion criteria in more detail.

- Describe procedures like endoscopy, pH/impedance monitoring, manometry, PPI trial. Provide relevant parameters.

- Provide clear definitions for all GERD phenotypes and criteria used for classification.

- Explain how MNBI and PSPW were measured and calculated from pH/impedance data.

- Elaborate on statistical tests used for data analysis.

Results

- Provide a table summarizing patient demographics, clinical characteristics, and key outcomes for each GERD phenotype.

- Present MNBI and PSPW values for each phenotype in box plots or dot plots to showcase the distributions.

- Report statistical values - p-values, confidence intervals etc - for comparisons between phenotypes.

- Include a table detailing the performance of MNBI and PSPW in distinguishing phenotypes - AUC, sensitivities, specificities.

- Show the ROC curve for PSPW predicting pathological MNBI.

Discussion

- Elaborate on the potential mechanisms explaining the strong correlation found between MNBI and PSPW.

- Discuss how dilated intercellular spaces could underlie changes seen in both parameters.

- Compare your MNBI and PSPW values across phenotypes to reference ranges reported in literature.

- Discuss factors that could affect MNBI and PSPW values like geographical differences, measuring devices etc.

- Suggest future research directions - prospective studies, automated measurement, implications for diagnosis and treatment, possibly related.

- Highlight strengths of your study like simultaneous evaluation of both parameters. Discuss possible comorbidities .

Comments on the Quality of English Language

none

Author Response

Thank you very much for the comments and recommendations. We have addressed your concern and make the proper corrections.

Introduction

  1. Provide more details on the prevalence and burden of GERD globally and LPR. Cite key statistics.

More details have been provided for the prevalence and burden of GERD globally based on the two largest studies available at this moment in literature.

Concerning LPR: Despite the recognition of LPR as a distinct condition over three decades ago, there is a lack of standardized diagnostic and therapeutic guidelines. This has resulted in frequent misdiagnosis of LPR in primary care settings. Additionally, the prevalence and contributing factors of LPR within the local community remain unknown. The absence of standardized diagnostic criteria has hindered the development of accurate prevalence and burden studies for LPR. Furthermore, considering GERD and LPR are distinct entities, we do not consider it being relevant to our study. 

  1. Elaborate on the different phenotypes of GERD - erosive reflux disease (ERD), non-erosive reflux disease (NERD), functional heartburn (FH), reflux hypersensitivity (RH).

We have included further details on GERD phenotypes in the introduction.

  1. Explain the pathophysiology behind altered esophageal mucosal integrity in GERD. Discuss dilated intercellular spaces (DIS).

We have added further pathophysiology explanations and discussed DIS.

  1. Introduce multichannel intraluminal impedance pH monitoring (MII-pH). Define key parameters like mean nocturnal baseline impedance (MNBI) and post-reflux swallow-induced peristaltic wave index (PSPW).

We have added new information to clearly define parameters in the introduction section

  1. Discuss existing literature on the utility of MNBI and PSPW in diagnosing GERD phenotypes.

This issue is elaborated in the discussion section in order to provide better cohesion and accessibility to the manuscript.

  1. State the rationale and objectives of your study clearly.

Rephrased the text to be clearer.

Methods

7. Provide details on study design, setting, timeline and ethical approvals obtained.

We have provided details concerning these issues

8. Explain patient recruitment strategy and inclusion/exclusion criteria in more detail.

More detail was added in the text

9.  Describe procedures like endoscopy, pH/impedance monitoring, manometry, PPI trial. Provide relevant parameters.

Procedures were described and relevant parameters were added.

10.  Provide clear definitions for all GERD phenotypes and criteria used for classification.

Clear definitions were added.

11. Explain how MNBI and PSPW were measured and calculated from pH/impedance data.

The way MNBI and PSPW were measured and calculated from pH/impedance data are presented in the text.

12. Elaborate on statistical tests used for data analysis.

Further details were added.

Results

13.  Provide a table summarizing patient demographics, clinical characteristics, and key outcomes for each GERD phenotype.

Table 1 provides these details

14.  Present MNBI and PSPW values for each phenotype in box plots or dot plots to showcase the distributions.

Boxplots for each phenotype for MNBI and PSPW values are presented in Figure 1 and 2.

15.  Report statistical values - p-values, confidence intervals etc - for comparisons between phenotypes.

P values and confidence intervals were added were added in Table 2 where there is comparisons between phenotypes

16.  Include a table detailing the performance of MNBI and PSPW in distinguishing phenotypes - AUC, sensitivities, specificities.

This data is presented in Table 2.

17.  Show the ROC curve for PSPW predicting pathological MNBI.

Figure 3 shows the ROC curve for PSPW predicting pathological MNBI

Discussion

18. Elaborate on the potential mechanisms explaining the strong correlation found between MNBI and PSPW.

We added more information on this topic.

  1. Discuss how dilated intercellular spaces could underlie changes seen in both parameters.

We added this in the discussion section.

20. Compare your MNBI and PSPW values across phenotypes to reference ranges reported in literature.

This issue was addresses in the discussion section. MNBI and PSPW values vary widely across geographic locations, with large discrepancies between what is considered normal in different regions of the world, making reference ranges difficult to ascertain.

21. Discuss factors that could affect MNBI and PSPW values like geographical differences, measuring devices etc.

More factors were added in the discussion section.

  1. Suggest future research directions - prospective studies, automated measurement, implications for diagnosis and treatment, possibly related.

We added future research directions such as Artificial Intelligence and automated measurements.

23. Highlight strengths of your study like simultaneous evaluation of both parameters. Discuss possible comorbidities.

Strengths of the study were added. There were no comorbidities that influenced results in our study and BMI as a parameter was not useful in differentiating GERD phenotypes.

Reviewer 2 Report

Comments and Suggestions for Authors

This study is interesting because it confirms the diagnostic value of the two new impedance metrics proposed in the diagnosis of GERD. The most important limitations are represented by its retrospective design and the small sample sizes of the GERD phenotypes the Authors analysed. They could influence some of the results obtained in the study. Other specific comments are the following:

- The presence of DIS is a sensitive marker of microscopic mucosal damage and two of the most important studies in this field should be quoted in the references (Zentilin P et al, Am J Gastroenterol 2005, 100:2299 and Savarino E et al, J Gastroenterol 2013, 48:473).

- The definitions of NERD and RH do not require the response to PPIs (see Rome IV criteria)

- The authors should emphasize in the discussion that the new impedance metrics are so useful that they overcome the important limitations of traditional indices of symptom-reflux association (SI and SAP), which should be given up

Comments on the Quality of English Language

There are few grammatical errors (e.g. facility instead of facilitate)

Author Response

Thank you very much for the comments and recommendations. We have addressed your concern and make the proper corrections.

This study is interesting because it confirms the diagnostic value of the two new impedance metrics proposed in the diagnosis of GERD. The most important limitations are represented by its retrospective design and the small sample sizes of the GERD phenotypes the Authors analysed. They could influence some of the results obtained in the study. Other specific comments are the following:

  1. The presence of DIS is a sensitive marker of microscopic mucosal damage and two of the most important studies in this field should be quoted in the references (Zentilin P et al, Am J Gastroenterol 2005, 100:2299 and Savarino E et al, J Gastroenterol 2013, 48:473).

We added the references and expanded that section.

  1. The definitions of NERD and RH do not require the response to PPIs (see Rome IV criteria)

Thank you for pointing that out, we have modified the manuscript.  

  1. The authors should emphasize in the discussion that the new impedance metrics are so useful that they overcome the important limitations of traditional indices of symptom-reflux association (SI and SAP), which should be given up

Thank you for your input. We added more information in the manuscript to address this issue.